# Communication of an Abnormal Metabolic New-Born Screening Result in The Netherlands: The Parental Perspective

**DOI:** 10.3390/nu14193961

**Published:** 2022-09-23

**Authors:** Sietske Haitjema, Charlotte M. A. Lubout, Justine H. M. Zijlstra, Bruce H. R. Wolffenbuttel, Francjan J. van Spronsen

**Affiliations:** 1Department of Metabolic Diseases, Beatrix Children’s Hospital, University of Groningen, University Medical Center Groningen, 9700 RB Groningen, The Netherlands; 2Department of Endocrinology, University of Groningen, University Medical Center Groningen, 9700 RB Groningen, The Netherlands

**Keywords:** new-born screening, phenylketonuria, communication, parental perspective, general practitioner

## Abstract

In the Netherlands, abnormal New-Born Screening (NBS) results are communicated to parents by the general practitioner (GP). Good communication and consequential trust in professionals is of the utmost importance in the treatment of phenylketonuria (PKU). The aim of this study was to assess parental satisfaction regarding the communication of an abnormal NBS result for PKU in the Netherlands. An email containing the link to a web-based questionnaire was sent by the Dutch PKU Association to their members. Responses to open questions were categorized, data of both open and closed questions were analysed with descriptive statistics and the Chi-Square test using SPSS. Out of 113 parents of a child with PKU (born between 1979 and 2020), 68 stated they were overall unsatisfied with the first communication of the NBS result. Seventy-five parents indicated that wrong or no information about PKU was given. A significant decrease was found in the number of parents being contact by their own GP over the course of 40 years (*p* < 0.05). More than half of all parents were overall unsatisfied with the first communication of the abnormal NBS result for PKU. Further research on how to optimize communication of an abnormal NBS results is necessary.

## 1. Introduction

New-Born Screening (NBS) for inherited metabolic disorders enables an early start of treatment for specific metabolic diseases and helps to prevent severe sequelae [1,2]. The procedure of the first communication of an abnormal NBS result to parents varies from country to country [3,4]. In the Netherlands, results are almost always communicated to the parents by their general practitioner (GP) [5,6]. 

Receiving an abnormal NBS result can be seen as life-altering news. It not only influences short-term parental outcomes [7,8,9] but can also have a long-term impact on both children and parents [10,11]. An abnormal NBS result can, for example, lead to parental stress and anxiety, hampering optimal treatment adherence. Therefore, good communication is crucial for caregiver’s look on the disease and its treatment [2,12,13]. 

Results from earlier studies demonstrate the importance of uniformed guidelines on how to communicate an abnormal NBS result [2,12,13,14]. For example, Chudleigh et al. [14] have recently evaluated interventions, designed by parents and health care professionals together, which could help standardizing communication of an abnormal NBS result in the United Kingdom. In the Netherlands, the need for a uniform follow-up protocol has been emphasized earlier by Blom et al. [13]. GPs currently receive a written leaflet with disease information in laymen’s terms when they obtain the NBS result from the medical advisor of the National Institute for Public Health and the Environment (RIVM) [15]. There is, however, no uniform protocol or material on how to give well-balanced information.

Most of the above mentioned studies have been performed in cystic fibrosis and/or sickle cell disease but not in metabolic diseases leading to an urgent referral. The urgency of the referral might have an impact on the level of stress and anxiety experienced by parents. Phenylketonuria (PKU) is a good metabolic example for investigating parental satisfaction regarding the communication of an abnormal NBS result, as the number of patients is relatively large. The NBS for PKU on a national level was introduced in 1974 after a pilot in the northern part of The Netherlands in 1969 [16,17]. 

PKU (OMIM #261600) is an inborn error of metabolism caused by mutations in the phenylalanine hydroxylase (PAH) gene. PAH converts phenylalanine (Phe) to tyrosine (Tyr), facilitated by the co-substrate tetrahydrobiopterin (BH4). Untreated PKU leads to Phe accumulation that causes severe intellectual disability, epilepsy and behavioural problems. The cornerstone of treatment is dietary phenylalanine restriction [18]. Early start of this dietary treatment is of paramount importance since any delay may have an impact on above mentioned consequences. Adherence to this treatment requires parental trust in professionals from the beginning. 

We hypothesized that there might be parental unsatisfaction about the communication of the NBS result of PKU arising due to the following three elements: (1) GPs need to break the bad news to a family with a healthy looking new-born, not expecting anything; (2) GPs have to refer them instantly to a university medical centre; (3) the diseases screened for by NBS are often very rare, and GPs may have never heard of them before. In order to improve the communication of an abnormal NBS result it is important to be aware of the common pitfalls experienced by parents receiving the abnormal result. This study therefore aimed to assess parental satisfaction regarding the communication of an abnormal NBS result for PKU. 

## 2. Materials and Methods

### 2.1. Newborn Screening Protocol in The Netherlands 

The steps currently taken in the NBS in the Netherlands are as follows. A blood spot is collected from the heel between 72 and 168 h after birth by a trained nurse. These samples are sent to the regional screening laboratory. Abnormal results are first reported to the medical advisor of the RIVM during weekdays, who will contact the regional metabolic paediatrician of one of the 6 university medical centres. The GP is contacted by the medical advisor thereafter, usually in the late afternoon. The medical advisor asks the GP to visit the family at home to see the child, to communicate the abnormal result, and to refer to the metabolic paediatrician. The metabolic paediatrician sees the child with an abnormal screening result that evening or the next morning depending on various factors [5].

### 2.2. Questionnaire Development and Distribution 

This study was initiated by the Dutch PKU Association (DPKUA) for patients and parents. All members of the DPKUA were approached by e-mail. They were asked to fill in a Dutch open web-based questionnaire on parental satisfaction regarding the first contact with the GP about the abnormal NBS results, if they have had experience with this (e.g., being a parent and/or caregiver of a child with PKU). The questionnaire was developed in close cooperation with the DPKUA and a small group of parents, and consisted of 7 closed questions and 8 open-ended questions. The final questionnaire comprised 15 questions (Appendix A) and was sent out in February 2021 without recall. In consultation with the Medical Ethical Committee of Groningen, no waiver was needed.

### 2.3. Patient and Public Involvement

Members (caregivers of patients) of the DPKUA were involved in the design and conduct of this research. One focus group session with members and the board of the DPKUA was held and important topics were identified. Once the study has been published, participants will be informed of the results through the DPKUA. 

### 2.4. Data Analysis

An anonymized datafile generated by the DPKUA was used for analysis. After indicating whether parents were overall satisfied or overall unsatisfied, parents could still give reasons for both options. For the responses to open questions multiple categories per response were possible. Before categorization, all answers to the open-ended questions were evaluated in order to compile the most effective categories. To ensure high-quality analysis, all answers and categories were again checked by an independent person (JHMZ). The questionnaire was performed in Dutch, categories and answers given in the results section were translated and again checked by an independent person (JHMZ). Any discrepancy resulted in a discussion with JHMZ and SH until consensus was reached.

To assess trends over time two groups were defined: group 1 consisted of parents with a child diagnosed between 1979 and 2006; group 2 of parents with a child diagnosed between 2006 and 2020. Grouping was based on dividing participants in two almost equal groups in sample size. Data were analysed using descriptive statistics and the Chi-Square test for the differences between groups using IBM SPSS Statistics 23. 

## 3. Results

### 3.1. Participants

The questionnaire was sent to all 439 members of the DPKUA, 282 members opened the email and 113 of them completed the questionnaire. However, the administration of the DPKUA can not distinquish between parents and patients as members of the association. Of the 113 parents of children with PKU who completed the questionnaire, group 1 (born/diagnosed in 1979–2006) consisted of 59 parents, while group 2 (patient born/diagnosed in 2006–2020) consisted of 54 parents. 

### 3.2. Overall Satisfaction

In total, just under 40% of all parents indicated they were overall satisfied with the way the abnormal NBS result was communicated to them (Table 1). Of the 113 parents, 69.9% was informed by their own GP, 13.3% by a replacing GP, 4.4% by the GP’s assistant, and 12.4% by another health-care provider (e.g., paediatrician or midwife). 37.2% of the parents were contacted by phone, in 48.7% the GP performed a home visit, and in the other 14.1% parents were contacted by other means (e.g., voicemail, midwife at home, paediatrician/other doctor in the hospital). Overall satisfaction decreased over time, from 44.1% in group 1, to 35.2% in group 2, although not statistically significant. A statistically significant difference was found when comparing the two sub-groups: parents from group 1 were most often contacted by their own GP compared to group 2 (81.4% vs. 57.4%; *p* < 0.05).

Overall, just under 50% of the parents indicated the GP had contacted the metabolic paediatrician before their conversation, while 27.2% indicated the GP contacted the paediatrician during the visit to the parents, thereby performing a tandem conversation (Table 1). When looking at group 1 versus group 2, GPs in group 2 more often performed a tandem conversation with the paediatrician during the visit to the parents in comparison to group 1, 38.9% vs. 16.9% respectively (*p* < 0.05). Most parents stated they received clear instructions about where, at what time, and with whom they were expected in the hospital. In 25.7% of the cases, parents were reached out by or had contact with the GP to evaluate the communication of the NBS results afterwards. More than half of all parents indicated the reception in the hospital was good: ‘The reception in the hospital was very good, they understood the insecurity and fear of the parents very well’.

Section 3.3, Section 3.4 and Section 3.5 consisted of open-ended questions. Multiple categories per response were possible in every open-ended question. Table 2 shows all categories per question. 

### 3.3. Reasons behind (un)Satisfaction 

Of the 113 participants, 61 gave reason for their (un)satisfaction (Table 2). The most important reason to be satisfied was a home visit performed by the GP. Most participants were satisfied about multiple categories, for example, one participant said: ‘The GP first gave us a call, he was very clear and reassuring. He told us there was an abnormal NBS result, that we shouldn’t worry too much, that it was not life threatening, but that something needed to happen, and that he would be at our home in 10 min. It felt nice that he came to our home so quickly. He was very clear that he did not know much about PKU, that he did some research shortly before, but that they would help us further in the hospital.’ When comparing the reasons to be (un)satisfied of group 1 to the reasons of group 2, one significant difference was found between parents’ opinions about having an involved GP at that time, being 13.8% vs. 37.5% respectively (*p* < 0.05). 

The most important reason to be unsatisfied (Table 2) was that the GP was not able to give correct information or gave wrong information about PKU, for example: ‘It was unclear what exactly was going on, the GP clearly did not know PKU either. Because of the vagueness, ‘‘brain damage, directly to the hospital, don’t eat anything anymore’’, there is more panic than would have been necessary’.

### 3.4. Information Received from the General Practitioner

Table 2 also summarizes the GP’s main message, according to the memory of the parents. In most cases, parents were (among others) told that they were expected in the hospital with some urgency. As one participant said: ‘The GP only told us that it was PKU, that it was serious, and that we had to take our child to the hospital as soon as possible’.

### 3.5. Parental Suggestions on Optimizing the Communication of NBS Results

Possible improvements in the communication of the NBS results were suggested by 81 parents (Table 2). Of those, 50 suggested the explanation and awareness of PKU among GPs could be improved, while 33 indicated the GP should always perform a home visit: ‘I think a home visit from the GP is very important since you get a message which directly turns your life as a new parent upside down’.

## 4. Discussion

This study is, to the best of our knowledge, the first to assess the parental perception on communicating an abnormal NBS result of a metabolic disease by the GP in the Netherlands. The most important finding is that more than half of all participants stated they were overall unsatisfied with the way the abnormal NBS result was communicated. 

The findings in this report are subject to at least four limitations. First of all, as a limitation of a questionnaire, questions and answers may have been interpreted wrongly. In this study, we only assessed the parental satisfaction and their look on the GPs’ role in the NBS procedures. We did not involve the GPs themselves in this study. Additionally, as members of the DPKUA were approached, we have only looked at true-positive screening results. False-positive results are, on the other hand, known to cause increased parental stress, and parents might experience the communication differently or might be focused on different points in the communication of the NBS result [19]. Lastly, for some parents this event happened over 40 years ago. Therefore, there might be recall bias. 

The main reason given for the overall unsatisfaction was the receival of wrong information about PKU or even no information at all. This finding is consistent with that of previous reports regarding parents’ perspectives in severe combined immunodeficiency, cystic fibrosis and sickle cell disease [7,8,9,10,12]. The lack of correct information might increase anxiety and can lead to low confidence in treatment, which is most important in the early days after diagnosis [3,20]. The majority of parents was satisfied when the GP, who communicated the NBS result, visited them at home. This home visit gave parents the feeling that the GP was engaged, and made it possible for the GP to reassure the parents. On the contrary, parents who received the result by phone felt very distanced from the GP, especially when the message was given by the assistant. 

Parental unsatisfaction increased (not significantly) over the last 40 years. A possible explanation for this could be recall bias. However, it is shown that parents seem to remember this event in great detail, also underlining the impact of the event. The number of home visits performed by the GP decreased (not significantly) over time. It could be hypothesized that the decrease in home visits plays role in the decreased satisfaction over time. The reduction in home visits might be explained by the increasing workload of the GPs and the lack of time to perform home visits [21].

In general, GPs can be expected to have enough experience in breaking bad news, and they often know the families well [10,22,23,24]. It seems, however, reasonable that breaking bad news to parents not expecting anything is quite different breaking bad news to parents who are worried about their child [25]. Another factor contributing to the complexity of communicating an abnormal NBS result is the timing of the message: it is within the new-born period which can be a very intensive period for families. As one participant put it: ‘Due to my education I knew the consequences of untreated PKU, so when receiving this message I felt paralyzed. I also just went through a tough labour, which I all had to put aside to be with my daughter in the hospital’.

It is imaginable, especially in abnormal NBS results needing urgent care, that the GP does not have time beforehand to read through information about the possible disorder. In addition, the urgency of the referral can also possibly lead to an increase in parental anxiety. Previous studies on a less urgent disease screened for by NBS, such as CF, showed mixed results [26,27,28,29,30,31,32,33]. For example, Barben et al. [26] reported on the importance of information giving in the first contact, and suggested that the GP may be inadequately prepared to perform this role. On the contrary, several other studies showed a good overall satisfaction with the communication of an abnormal NBS result for CF, although not always being performed by the GP [27,28,29]. Further studies are needed to identify possible differences between communicating an urgent versus a less urgent NBS result. 

When asking parents what could be improved in the communication of the abnormal NBS result, the majority indicated that a better explanation of PKU was necessary. Since the NBS was initiated almost 50 years ago for diagnosing PKU [30], it could be hypothesized that over the course of time more information and awareness would have originated among GPs, facilitating communication. However, this has not resulted in a rise in satisfaction. Even though the existing Dutch act sheets from the National Institute for Public Health and the Environment contain additional information and actions that need to be undertaken, they do not contain information on how to communicate and balance the NBS results [15]. Hence, it could conceivably be hypothesized that the GP needs other information to communicate the NBS results. One could think of an instruction video or an adjusted info sheet. 

Another aspect that could help improve the communication of an abnormal NBS result, and especially the reassurance of parents, is consultation with the metabolic paediatrician before and/or during the conversation with the parents (e.g., a tandem phone call), as suggested earlier by Blom et al. [13]. Even though most parents indicated that the GP did not perform a tandem phone call during their conversation about the abnormal NBS result, the number of tandem phone calls increased significantly over time. Despite not directly leading to a rise in satisfaction in this study, it is thought that the GP and paediatrician together can give parents the necessary and correct information immediately, thereby covering both key elements of familiarity and knowledge. 

In addition, it could even be questioned whether giving disease-specific information should be done by someone other than the GP. In Switzerland, for example, the information provided during the first phone call for CF is deliberately minimal. To reduce parental anxiety and to discourage parents from performing a web search on CF, the disease itself is not mentioned until they see the specialist, where they receive accurate information [31]. Another example is the UK, where the first communication is often done by a specialized nurse [32]. Even though the specialized nurse is often not someone familiar to the family, it is perhaps better to have someone who has the knowledge to provide disease-specific information but who is not known to the family than vice versa. A different route could be the metabolic paediatrician performing the first communication. However, this person is again often not familiar to the family and is not able to perform a home visit as the GP does due to distance to the family. 

To improve communication of abnormal NBS results, it is important to involve health professionals’ experiences [25,33]. Chudleigh et al. [14] aimed to implement and evaluate co-designed interventions that consisted of standardized laboratory proformas, communication checklists, and an email/letter template. They stated that some clinicians thought standardization of the first communication would be beneficial, while others felt this was not always possible due to the individualized needs of parents. The results of this study can further help define the difficulties and improve communication of the NBS result to parents by GPs. 

Feeling at ease with and accepting the disease is one of the most important factors influencing metabolic control of the child. Feeling at ease is influenced by, amongst others, the way of communicating the results [34]. A poor communication may negatively influence parents’ perception of information given by the professionals afterwards. In addition, lack of good communication and information may increase the risk of a posttraumatic stress syndrome, as is seen in several cases with parents of children with other chronic conditions [35]. Good communication should, therefore, be fair, transparent, and comfortable [36]. 

In conclusion, this study set out to explore parental satisfaction in the communication of an abnormal metabolic NBS result, showing overall low satisfaction and proposing several steps to improve the communication of an abnormal NBS result.

## Figures and Tables

**Table 1 nutrients-14-03961-t001:** Summary of closed ended questions.

**Categories**	**Overall (*n* = 113)**
Overall Satisfied	45 (39.8%)
Contact GP with metabolic paediatrician before conversation—yes	53 (46.9%)
Contact GP with metabolic paediatrician during conversation (tandem conversation)—yes	31 (27.2%)
Contact GP with parents after referral—yes	29 (25.7%)
Good reception in the hospital—yes	90 (79.6%)
Clear instructions about where to go (e.g., hospital, ward, etc.)—yes	77 (68.1%)

**Table 2 nutrients-14-03961-t002:** Summary of overall results per category per question.

**Categories per Question**	**Overall**
**Satisfied**	*n* = 61
Performed a home visit	39 (63.9%)
Involved GP	16 (26.2%)
Quick referral to hospital	14 (23%)
Honesty about own experience	10 (16.4%)
Gave correct information	8 (13.1%)
Quick diagnosis	4 (6.6%)
**Unsatisfied**	*n* = 79
Gave wrong/no information about PKU	57 (72.2%)
Send to the hospital without information	26 (32.9%)
Diagnosis via phone	16 (20.3%)
Unfamiliar GP	10 (12.7%)
Impersonal	9 (11.4%)
Home alone during the visit	2 (2.5%)
**Message received from the GP**	*n* = 106
Expected in the hospital (with some urgency)	71 (67.0%)
Diagnosis PKU/disorder in metabolism	45 (42.1%)
There is something wrong with the results from the NBS	42 (39.6%)
Nothing/almost no explanation	21 (19.6%)
Treatable with diet and special foods	11 (10.3%)
Explained what PKU is	7 (6.5%)
I don’t remember	5 (4.7%)
You need to count on a stay of a few nights	5 (4.7%)
Severe brain damage	2 (1.9%)
Not life-threatening	2 (1.9%)
**Parental suggestions on optimizing the communication**	*n* = 81
Clear explanation of PKU	50 (61.7%)
Personal visit/sufficient time from the GP	33 (40.7%)
Bring parents in contact with other parents of children with PKU	4 (4.9%)
Message from specialized healthcare workers instead of GP	3 (3.7%)
Protocol for GP how to handle abnormal NBS results	3 (3.7%)
Always receive a message, despite the NBS outcome	3 (3.7%)

## Data Availability

Data (e.g., questionnaire and questionnaire results) are available upon request.

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
