# Peer review of "Communication of an Abnormal Metabolic New-Born Screening Result in The Netherlands: The Parental Perspective"

_nutrients, 2022, doi:10.3390/nu14193961_

Round 1

Reviewer 1 Report

I would be interested in:

1. Comments about the practise of a positive screen result found on a Friday, or a weekend day - this is really important to acknowledge.

2.  Is there any significance in the grouping of respondents 1979-2006 and 2006-2020 - what happened in the middle of 2006 - did anything happen relevant to this.  (eg "Dr Google" reaches the Netherlands - I think it was before that year).

3. Were the authors not at all interested in any 2nd positive screen result experienced by the parents, ie for a younger sibling with PKU? I suppose not.

4. Ref 27 is mislabelled 31 in the text - line 243

5. Line 251 - sixty years - seems to be different to the date NBS actually started throughout the Netherlands - seen in line 55 as 1974

I like this paper very much, an excellent topic. Great paper.

Reviewer 2 Report

This work aims to evaluate the quality of communication to parents of a positive result in NBS for PKU from the parents perspective in the Netherlands between 1979 and 2020. The study concludes that most of the participants were unsatisfied with the way in which the result was communicated to them, and analyzes the possible causes and needs for improvement.  

Although this study has important limitations -recognized by the authors-, it is methodologically correct and provides valuable information for the future of this and other NBS programs.

MINOR CHANGES

Line 114. I suggest explaining based on what criteria these periods (1979-2006 and 2006-2020) have been selected. Was there any relevant change over the years in how communication with parents was carried out?

Discussion. Although the tandem phone call option is considered, it is not discussed the possibility that the first communication to the family could be made directly by the metabolic paediatrician (who knows well the disease) as it is done by some NBS programs in Europe. Since many parents complained that they have received wrong information about PKU or even no information at all, this option should be considered and discussed in this section.
